# Inhibition of *Acinetobacter baumannii* Biofilm Formation by Terpenes from Oregano (*Lippia graveolens*) Essential Oil

**DOI:** 10.3390/antibiotics12101539

**Published:** 2023-10-14

**Authors:** Melvin Roberto Tapia-Rodriguez, Ernesto Uriel Cantu-Soto, Francisco Javier Vazquez-Armenta, Ariadna Thalia Bernal-Mercado, Jesus Fernando Ayala-Zavala

**Affiliations:** 1Departamento de Biotecnología y Ciencias Alimentarias, Instituto Tecnológico de Sonora, 5 de Febrero 818 Sur, Col. Centro, Ciudad Obregón 85000, Mexico; ernesto.cantu18818@potros.itson.edu.mx; 2Departamento de Ciencias Químico Biológicas, Universidad de Sonora, México Blvd. Luis Encinas y Rosales S/N, Col. Centro, Hermosillo 83000, Mexico; franciscojavier.vazquez@unison.mx; 3Departamento de Investigación y Posgrado en Alimentos, Universidad de Sonora, México Blvd. Luis Encinas y Rosales S/N, Col. Centro, Hermosillo 83000, Mexico; thalia.bernal@unison.mx; 4Centro de Investigación en Alimentación y Desarrollo, A.C. Carretera Gustavo Enrique Astiazarán Rosas 46, Hermosillo 83304, Mexico; jayala@ciad.mx

**Keywords:** biofilm formation, terpene compounds, carvacrol, oregano essential oil, thymol, antimicrobial activity, twitching motility, antibiotic resistance

## Abstract

*Acinetobacter baumannii* is a nosocomial pathogen known for its ability to form biofilms, leading to persistent infections and antibiotic resistance. The limited effective antibiotics have encouraged the development of innovative strategies such as using essential oils and their constituents. This study evaluated the efficacy of oregano (*Lippia graveolens*) essential oil (OEO) and its terpene compounds, carvacrol and thymol, in inhibiting *A. baumannii* biofilms. These treatments showed a minimum inhibitory concentration of 0.6, 0.3, and 2.5 mg/mL and a minimum bactericidal concentration of 1.2, 0.6, and 5 mg/mL, respectively. Sub-inhibitory doses of each treatment and the OEO significantly reduced biofilm biomass and the covered area of *A. baumannii* biofilms as measured by fluorescence microscopy. Carvacrol at 0.15 mg/mL exhibited the most potent efficacy, achieving a remarkable 95% reduction. Sub-inhibitory concentrations of carvacrol significantly reduced the biofilm formation of *A. baumannii* in stainless steel surfaces by up to 1.15 log CFU/cm^2^ compared to untreated bacteria. The OEO and thymol exhibited reductions of 0.6 log CFU/cm^2^ and 0.4 log CFU/cm^2^, respectively, without affecting cell viability. Moreover, the terpenes inhibited twitching motility, a crucial step in biofilm establishment, with carvacrol exhibiting the highest inhibition, followed by OEO and thymol. The study provides valuable insights into the potential of terpenes as effective agents against *A. baumannii* biofilms, offering promising avenues for developing novel strategies to prevent persistent infections and overcome antibiotic resistance.

## 1. Introduction

*Acinetobacter baumannii* is an opportunistic Gram-negative bacterium that has emerged as the leading cause of nosocomial infections due to its multidrug resistance [1]. This pathogen is the first on the critical priority list for novel antibiotic development and poses a significant public health risk [2,3]. The Center for Disease Control and Prevention (CDC) have highlighted the increasing incidence and prevalence of *A. baumannii* infections in healthcare settings worldwide [4]. Various physiological processes, including efflux pumps and specific mutations, cause antimicrobial resistance. Nevertheless, biofilms impact substantially on this phenomenon. Biofilm resistance is attributed to the physical encapsulation of cells, isolation from the environment, and phenotypic differentiation within the biofilm, which makes some cells metabolically slower and less sensitive to antimicrobials [5]. The ability of *A. baumannii* to form biofilm on biotic and abiotic surfaces has contributed to the majority of nosocomial infections, most typically biofilm-associated illnesses such as ventilator-associated pneumonia and catheter-related infections [6]. These biofilms allow the pathogen to resist antibiotics more, cause persistent infections, and thrive in a hospital environment, spreading antibiotic-resistant strains and increasing hospital stays and healthcare costs [7,8].

Biofilms are structured bacterial communities that adhere to surfaces with a different metabolism than planktonic cells. These communities are encased within a self-produced matrix of extracellular polymeric substances that protect against antibiotic action, immune response, and environmental stress [3]. Bacterial biofilm development is a complex process involving many secreted proteins, extracellular appendages, and quorum-sensing systems. For example, type IV pili are multiprotein bacterial surface appendages assembled by *A. baumannii* [9]. These pili can be rapidly extended or retracted, participating in natural transformation, twitching motility, and surface adhesion [10]. These pili stick to solid surfaces and allow bacteria to travel on those surfaces, making initial colonization more efficient, which is essential for biofilm formation [5]. Therefore, the inhibition of this pili could be seen as a strategy to counteract *A. baumannii* biofilm formation.

*A. baumannii*-related biofilm infections can be exceedingly resistant to antibiotic therapy, making clinical management more complicated [11]. The current situation necessitates developing novel therapeutic procedures to ensure successful treatment outcomes. As traditional therapeutic approaches are ineffective, alternative strategies utilizing natural compounds with antimicrobial potential have gained considerable attention [12]. Among these natural compounds, oregano essential oil (OEO) has emerged as a promising candidate due to its complex composition rich in terpenes such as carvacrol, thymol, p-cymene, γ-terpinene, terpinen-4-ol, and linalool [13]. Among these, carvacrol and timol are two antibacterial compounds that have shown significant activity against a wide range of microorganisms due to the presence of a hydroxyl group in their structure [14]. Carvacrol and thymol also can be found in the EOs of other medicinal plants such as thyme (*Thymus vulgaris*), savory (*Satureja hortensis*), *Thymbra spicata*, and *Zataria multiflora* [15,16].

In recent years, research has shown that these terpenes exhibit inhibitory effects on biofilm formation via various pathogens, hinting at their potential application in combating biofilm-related infections [17]. For example, it was demonstrated that carvacrol can inhibit the biofilm formation of *Pseudomonas aeruginosa*, a respiratory pathogenic bacteria associated with multidrug-resistant infections [18]. Additionally, this terpene affects other virulence factors associated with biofilm formation, such as motility and quorum sensing [19]. Although *P. aeruginosa* and *A. baumannii* have marked distinctions in their genetic makeup, virulence profiles, and the spectrum of infections they provoke, they share common traits, including their propensity to cause nosocomial infections, their capacity for biofilm development and their resistance against numerous antibiotic classes. With this in mind, this study postulates that oregano essential oil and its terpenes carvacrol and thymol could also be attractive candidates for developing new therapies that can overcome antibiotic resistance mechanisms, notably the biofilm formation exhibited by *A. baumannii*. The antibiofilm activity of essential oils and their constituents are rarely studied in *A. baumannii*. This study aimed to investigate the inhibitory effect of OEO terpenes on *A. baumannii* biofilm development, to test terpenes responsible for the activity, and to elucidate their possible mode of action.

## 2. Results and Discussion

### 2.1. Minimum Inhibitory and Bactericidal Concentration

This study evaluated two of the main compounds in oregano essential oil: carvacrol and thymol. Although p-cymene is the second most abundant constituent in our OEO, the choice was made to use thymol and carvacrol because they are well-known compounds with documented antimicrobial properties. It has been observed that monoterpenes with oxygenated groups exhibit greater inhibitory activity against bacteria compared to hydrocarbon monoterpenes like p-cymene. However, in future studies, it would be interesting to assess the effect of p-cymene on the growth of bacteria in both planktonic and biofilm states of *A. baumannii.*

Table 1 shows the antibacterial activity of OEO, carvacrol, and thymol against *A. baumannii*. The tested doses exhibited remarkable efficacy in inhibiting bacterial growth, with MIC values of 0.6, 0.3, and 2.5 mg/mL for OEO, carvacrol, and thymol, respectively. Moreover, the determined MBCs were 1.2, 0.6, and 5 mg/mL for OEO, carvacrol, and thymol. These findings show the antimicrobial properties of these compounds against *A. baumannii*, suggesting their potential as effective agents in combating pathogenic infections. Minimum inhibitory concentrations (MIC) and minimum bactericidal concentrations (MBC) are fundamental parameters in bacterial pharmacophysiology. MIC denotes the lowest concentration of an antimicrobial agent capable of retarding bacterial growth without necessarily causing the demise of all present bacteria. Conversely, MBC indicates the lowest concentration of an antimicrobial agent that can eliminate every bacterium within a given population. In this scenario, no bacterium survives exposure to the antimicrobial. The critical difference between MIC and MBC is their effect on bacteria, with MIC halting growth and MBC decisively eliminating bacterial populations. This comprehension is imperative in determining whether an antimicrobial is bacteriostatic (arrests bacterial growth) or bactericidal (lethal to bacteria). Understanding both MIC and MBC is crucial for optimizing antibiotic or antimicrobial therapy, preventing the emergence of antibiotic resistance, and selecting the most appropriate antimicrobial based on the nature of the infection and the patient’s condition.

In this study, carvacrol exhibited the lowest MIC, with an MBC value that was twice the concentration needed to inhibit bacterial growth. This potent antimicrobial effect can be attributed to the reactivity of carvacrol and its ability to degrade membrane proteins and enhance cell permeability in a concentration-dependent manner [20]. The antibacterial mechanism of oregano terpenes, including carvacrol and thymol, against pathogenic bacteria involves the disruption of embedded membrane proteins, destabilization of cell wall lipids, inhibition of DNA synthesis, interference with enzymatic activity, and suppression of efflux pumps [20]. Previous research has documented carvacrol’s antibacterial potential against the respiratory pathogenic bacteria *P. aeruginosa*, showing MIC and MBC values of 1.2 to 2.4 mg/mL, respectively [18]. On the other hand, Choudhary et al. [21] evaluated the antibacterial properties of natural compounds, encompassing flavonoids, phenolic acids, organic acids, and terpenes, against the clinical isolates of *A. baumannii*. Within the tested compounds, the volatile compounds eugenol and geraniol outperformed the others, exhibiting larger inhibition zones in the agar diffusion assay and MIC values of 1 and 0.5 mg/mL, respectively.

Similarly, Alves et al. [22] determined the MICs and MBCs of the terpenes geranyl acetate, linalool, p-cymene, and *α*-pinene, the main constituents of coriander (*Coriandrum sativum*) essential oil, against five *A. baumannii* strains. Their findings showed that MIC values ranged from 2 to 256 µg/mL, with linalool as the compound that presented the lower MICs (2–8 µg/mL). Interestingly, the MBC values were the same as the MIC, evidencing the bactericidal nature of the antibacterial activity. The differences in the susceptibility of *A. baumannii* strains towards terpene compounds could be attributed to virulence factors associated with resistance, such as efflux pumps and beta-lactamase activity mechanisms. On the other hand, the outcomes of the present study contribute to the expanding comprehension of plant terpenes’ effectiveness in combatting respiratory pathogenic bacteria.

### 2.2. Changes in Biomass Production in A. baumannii Biofilms

To further investigate the inhibitory potential of OEO, carvacrol, and thymol against the *A. baumannii* biofilms, the fractions of the MIC values were used to evaluate their effect on total biomass (Figure 1). Specifically, 0.5, 0.25, and 0.125 × MIC fractions were chosen to investigate the dose-dependency of treatments. The biomass produced by *A. baumannii* biofilms formed without treatments (control group) was considered 100%. According to Figure 1, biofilms formed in the presence of OEO or its terpene constituents, carvacrol, and thymol, presented less biomass than the control group (*p* < 0.05), and this effect was dose-dependent. At the lowest dose (0.125 × MIC), OEO caused a biomass reduction of 28%, whereas thymol and carvacrol led to 21% and 15% less biomass than the control, respectively. As the concentration increased to 0.25 × and 0.5 × MIC, the percentage of biofilm reduction was more pronounced across all treatments. The highest biofilm reduction was observed at the MIC concentration, where OEO, thymol, and carvacrol resulted in reductions of 64%, 57%, and 66%, respectively. These findings highlight the efficacy of these compounds in impairing biofilm formation, with higher concentrations leading to more significant reductions in total biofilm biomass. 

The effectiveness of OEO, carvacrol, and thymol against *A. baumannii* can be compared to their efficacy against other pathogenic bacteria. Several studies investigated the antibiofilm properties of these compounds against various bacterial species [20]. In the case of carvacrol, its antibiofilm potential has been widely reported against antibiotics-resistant pathogenic bacteria [18]. This terpene shows a biomass reduction of 57% in *P. aeruginosa* biofilms at doses of 1.2 mg/mL. Comparing these values to the results obtained for *A. baumannii* in this study, carvacrol exhibits similar efficacy against both pathogens. Although direct comparisons to *A. baumannii* are limited, terpene compounds have demonstrated broad-spectrum activity against various pathogenic bacteria, underscoring their potential as a valuable antimicrobial agent.

### 2.3. Changes in Cell Density of A. baumannii Biofilms Formed in the Presence of OEO, Carvacrol, and Thymol

To gain insight into the anti-biofilm activity of OEO, carvacrol, and thymol, their effect on the initial attachment of *A. baumannii* to stainless steel surfaces and further biofilm development was evaluated using cell density quantification (log CFU/cm^2^) for 24 h (Figure 2). For this experiment, the sub-inhibitory dose of 0.5 × MIC was chosen to maintain cell viability during biofilm formation. As shown in Figure 2, *A. baumanii* cells attach to stainless steel surfaces in the first hours of incubation, steadily increasing over time to reach the highest level at 24 h (6.7 log CFU/cm^2^). At the first 3 h, there were no differences (*p* > 0.05) in the number of attached cells between the control and treatments. Nevertheless, after 24 h of development, the biofilms formed in the presence of carvacrol, OEO, and thymol presented lower (*p* < 0.05) cell densities. Specifically, carvacrol treatment resulted in a significant reduction in cell density of biofilms of up to 1.15 log CFU/cm^2^ at 24 h compared to the untreated bacteria (*p* < 0.05). On the other hand, OEO and thymol caused reductions of 0.6 log CFU/cm^2^ and 0.4 log CFU/cm^2^, respectively. 

*A. baumannii* biofilm development involves sequential stages; the first steps are reversible and irreversible attachment to surfaces. Then, the attached bacteria grow onto the surface and establish microcolonies while producing extracellular polymeric substances (EPS) to form highly organized and resistant mature biofilms [22]. In addition, planktonic cells growing in the media continuously deposit onto layers of attached cells and contribute to increasing biofilm thickness during normal biofilm development [23]. Thus, understanding the whole process of biofilm formation could help to design more effective strategies to combat biofilm infection. In this sense, studies aimed to evaluate the effect of natural compounds on the biofilm development of *A. baumannii* are limited. In addition, those reported in the literature do not assess the effect of treatments on the different stages of biofilm development; instead, the anti-biofilm activity is determined at 24 h or 48 h of incubation [22,24]. Therefore, the results obtained in the present study show a broad picture of the action of natural terpenes at the early and advanced stages of *A. baumannii* biofilm formation.

In the present study, OEO, carvacrol, and thymol, at the evaluated concentration, did not affect the initial attachment of *A. baumannii* to stainless steel surfaces; however, mature biofilms (24 h) formed in the presence of these treatments showed lower cell densities. Since the experiments were performed at sub-inhibitory concentrations, the effects on bacterial growth can be discarded. Therefore, the lower cell densities of *A. baumannii* biofilm formed in the presence of treatments could be attributed to the impairment of cell-to-cell interactions that result from EPS production that helps hold bacterial populations together and shapes biofilm architecture [25]. This hypothesis is supported by the findings of Choudhary et al. [21], who reported that the volatile compounds eugenol and geraniol impaired the exopolysaccharide production, the main constituent of the extracellular matrix of *A. baumannii biofilms*, and downregulated the *csuE* gene, related to the synthesis of surface pili. Nevertheless, further studies should be carried out to determine if OEO, carvacrol, and thymol impair the EPS production during biofilm *A. baumannii* development.

### 2.4. Fluorescence Microscopy of A. baumannii Biofilms Exposed to OEO, Carvacrol and Thymol

In Figure 3, we examine the impact of sub-inhibitory doses, specifically 0.5 × MIC, of OEO, carvacrol, and thymol on *A. baumannii* biofilms using fluorescence microscopy. This assay enabled the visualization and assessment of structural changes induced by terpene compounds in biofilms. The aim was to elucidate the effects of these treatments on the colonization characteristics of *A. baumannii* biofilms. In this context, carvacrol, at a concentration of 0.15 mg/mL, exhibited remarkable efficacy by reducing the covered area of *A. baumannii* biofilms on glass surfaces by up to 95% (Figure 3a). Similarly, OEO (0.3 mg/mL) and thymol (1.25 mg/mL) treatments reduced the covered area by up to 65% and 16%, respectively. Microscopic analyses (Figure 3b) established that the application of terpene compounds led to a significant decrease in the covered area, indicating their ability to impact biofilm development regardless of the surface being tested.

Some studies have reported the effectivity of oregano essential oil against *A. baumannii* biofilm development; for example, it has been shown that nanosponges of OEO and carvacrol eradicate the biofilm formation of single and dual-species (*A. baumannii* and *S. aureus*) biofilms at 0.12 mg/mL and 0.24 mg/mL, respectively [26]. However, this report did not include changes in the covered area or other virulence factors as a target, as the present study did. Similarly, a recent study investigated the effect of chitosan-loaded nanoparticles with carvacrol on the biofilm formation of *A. baumannii* and reported a significant inhibition in the covered area of the biofilms at a dose of 0.15 mg/mL for 24 h [11]. These findings align with our results, suggesting that sub-inhibitory amounts of these terpene compounds, including carvacrol, could effectively modulate and damage the microcolony structures, reducing the colonization of *A. baumannii* biofilms. 

### 2.5. Inhibition of Twitching Motility

Despite the absence of flagellar motility in *A. baumannii*, targeting the disruption of Type IV pili-mediated twitching motility holds significant potential in impeding the establishment of stable biofilms [5]. For this reason, by reducing this key virulence factor, terpene compounds can be designed to hinder the initial attachment of *A. baumannii* to surfaces, thereby preventing the formation of strong biofilms. Figure 4 illustrates a substantial reduction in twitching motility observed in *A. baumannii* when exposed to OEO (0.3 mg/mL), carvacrol (0.15 mg/mL), and thymol (1.25 mg/mL). Remarkably, carvacrol exhibited the highest inhibition, with 13 mm in the twitching zone, compared to the 20 mm observed in untreated bacteria (*p* ≤ 0.05).

Many bacteria, including *Acinetobacter* and *Pseudomonas*, possess type IV pili, which are filamentous structures on their surfaces [27]. These pili are crucial in various bacterial functions such as motility and surface adhesion. The precise mechanism by which terpenes impact type IV pili remains unclear. However, drawing insights from previous research on other pathogens and anti-type IV pili agents, it is suggested that terpenes could exert their influence in two main ways. Terpenes can interfere with pili synthesis and assembly, hindering the bacterium’s ability to produce functional pili and directly disrupting existing type IV pili. This interference and disruption can lead to impaired motility and reduced surface adhesion, making it more challenging for the bacterium to form biofilms [5]. 

For instance, a recent study by Vo et al. [5] demonstrated the inhibitory effects of two phenothiazine compounds on twitching motility, which is dependent on type IV pili, and, consequently, on the formation of *Acinetobacter* biofilms. This research suggests that the mechanism of action of these compounds involves interference with pili biogenesis. Although these compounds do not have a structure like terpenes, similar effects have been reported to which these mechanisms can be attributed. For example, *Micromeria thymifolia* essential oil at 0.25 mg/mL affects *P. aeruginosa* twitching activity, showing 15 mm of diameter compared to 21 mm of control bacteria [28]. Indeed, this motility inhibition is attributed to terpenes limonene and terpinen-4-ol; these compounds reduce biofilm production and simultaneously inhibit the adhesion of *S. aureus* to polystyrene and glass surfaces [29]. 

It is important to note that the exact effects of terpenes on *A. baumannii* biofilm generation and pili type IV may vary depending on the terpene, its concentration, and the strain of *A. baumannii* studied. More research is needed to better understand terpenes’ processes and their potential applications in treating *A. baumannii* infections.

## 3. Materials and Methods

### 3.1. Bacteria Strains and Growth Conditions

*Acinetobacter baumannii* (ATCC 19606) was cultured aerobically in Luria Bertani broth (LB) and incubated at 37 °C for 24 h to obtain a bacterial density of 1 × 10^8^ CFU/mL.

### 3.2. Chemical Analysis of the Essential Oil

The oregano essential oil (*Lippia graveolens*) was obtained from ‘ORE aceite de orégano’ (Saucillo, Chihuahua, Mexico). The chemical composition of the oregano essential oil (OEO) was previously reported in our research group [30]. Notably, OEO primarily comprised carvacrol (47.4%) and thymol (3%). The terpenes carvacrol and thymol were purchased from Sigma Aldrich (Toluca, Mexico).

### 3.3. Minimal Inhibitory (MIC) and Bactericidal Concentrations (MBC) of OEO, Carvacrol and Thymol against A. baumannii

The micro-well dilution assay method was used to evaluate the antibacterial activity of OEO, carvacrol, and thymol against planktonic *A. baumannii* cells [18]. For this, 5 μL of bacteria suspension (1 × 10^8^ CFU/mL) were transferred to a 96-well polystyrene microplate (COSTAR) containing 295 μL of different concentrations (0–5 mg/mL) of the OEO and each compound dissolved in LB broth with 5% of DMSO. Microplates were incubated at 37 °C for 24 h. The MIC was determined as the lowest concentration that completely inhibits the visible growth of planktonic cells. To determine the MBC, the cultures from the wells that showed MIC, and the next three higher concentrations of OEO, thymol, and carvacrol were plated on LB agar and incubated at 37 °C for 24 h. The experiment was performed in triplicate, and the MIC and MBC were expressed as mg/mL. 

### 3.4. Biomass Production of A. baumannii Biofilms Exposed to OEO, Carvacrol, and Thymol

The inhibitory effect of OEO, carvacrol, and thymol against *A. baumannii*-attached biofilms was evaluated using a microtiter plate assay [18]. The test was performed by adding 5 μL of bacteria suspension (1 × 10^8^ CFU/mL) to a 96-well polystyrene microplate (COSTAR) followed by 295 μL of the OEO, and each compound at sub-inhibitory concentrations were dissolved in the LB broth using 5% of DMSO. The control was taken as the bacterial suspension without exposition to terpenes or OEO. Microplates were incubated at 37 °C for 24 h. Afterward, the cultures were discarded, and the plate was washed with sterile saline water to remove non-attached biofilms. The biomass attached to the polystyrene plate was stained with 300 μL of 0.1% crystal violet solution for 45 min to allow crystal violet to bind to components of the extracellular matrix and bacterial cells in the biofilm. The excess crystal violet solution was removed, the plate was washed with saline solution, and then it was allowed to dry. Then, 300 μL of 20% acetic acid was added to the microplate wells to solubilize the stained biomass. The released crystal violet is quantitatively measured via spectrophotometry (Fluostar Omega, BMG Labtech, Chicago, IL, USA) at 595 nm. The resulting absorbance correlates with the amount of biomass in the biofilm. The experiment was performed in triplicate, expressing the results as a percentage (%) of biomass production. 

### 3.5. Effect of OEO, Carvacrol, and Thymol on Cell Density of A. baumannii Biofilms during Its Development on Stainless Steel Surfaces

Stainless steel surfaces, including medical instruments, equipment, and hospital infrastructure, are commonly found in healthcare settings. The impact of OEO, carvacrol, and thymol on biofilm formation was assessed using stainless steel 304 coupons (1 × 1 × 0.1 mm). To achieve this, each coupon was immersed in 6 mL of LB broth and then inoculated to obtain a final concentration of 1 × 10^6^ CFU/mL of *A. baumannii*. Then, a dose of 0.5 × MIC of OEO, carvacrol, or thymol was added to these solutions, followed by incubation for 3, 6, 12, and 24 h at 37 °C. These sampling time points help to understand whether the treatments affect other aspects of biofilm formation, such as initial adhesion and colonization. The control group comprised bacterial solutions not exposed to terpenes or OEO. After each time, the coupons were removed, washed with sterile saline solution, and sonicated for 5 min in 3 mL of sterile saline solution. The adhered cells were quantified using serial dilutions plating in LB agar and incubated at 37 °C for 24 h. This experiment was conducted in triplicate, and the results were expressed as Log CFU/cm^2^.

### 3.6. Fluorescence Microscopy of A. baumannii Biofilms Exposed to OEO, Carvacrol and Thymol

Glass coverslips were utilized to facilitate fluorescence microscopy analysis, enabling the assessment of *A. baumannii* biofilms under sub-inhibitory concentrations of OEO, carvacrol, and thymol. These experiments were conducted using the same conditions as previously described. Each biofilm sample was stained with 0.1% Syto9, fixed for 30 min in a dark room, washed, and dried to observe at 200× magnification using an Axio Vert.A1 fluorescence microscope (Carl-Zeiss, White Plains, NY, USA). Moreover, the percentage of the covered area was calculated using ImageJ software version 2016 (KCL, London, UK) by measuring the rate of the covered area (%). This assay was repeated three times with three replicates per assay. 

### 3.7. Effect of OEO, Carvacrol, and Thymol on A. baumannii Twitching Motility

The twitching assay evaluated the effect of OEO, carvacrol, and thymol on *A. baumannii* IV pili activity associated with attachment in initial biofilm stages [31]. For this, an *A. baumannii* inoculum (1 × 10^6^ CFU/mL) was grown in LB broth in the presence of a dose of 0.5 × MIC of OEO, carvacrol, or thymol with 5% of DMSO at 37 °C for 24 h. The bacteria without exposure to the terpenes were taken as a control. After that time, 20 μL of the exposed bacteria were inoculated in the center of LB agar semisolid plates (0.5%) and incubated at 37 °C for 24 h. The adhesion spread was then measured after incubation time and expressed in mm. This experiment was performed in triplicate.

### 3.8. Statistical Analysis 

A completely randomized experimental design was employed for all assays. The effect of OEO and its terpenes at different concentrations (mg/mL) were examined for changes in *A. baumannii* biofilm biomass production (%), attached cells (log CFU/cm^2^), biofilm covered area (%), and twitching motility (mm). An analysis of variance (ANOVA) was conducted to determine significant differences, and a Tukey–Kramer test was performed for mean comparison. The level of significance was set at *p* ≤ 0.05 using the Number Cruncher Statistical Systems (NCSS) 2021 software.

## 4. Conclusions

The present study demonstrates the potent antimicrobial activity of oregano essential oil (OEO) and its terpene compounds, carvacrol, and thymol against *Acinetobacter baumannii* biofilms. The sub-inhibitory doses of these compounds effectively reduced biofilm formation and twitching motility, highlighting the disruption of Type IV pili-mediated mechanisms as a critical factor in impeding biofilm development. Carvacrol exhibited the highest efficacy in inhibiting biofilm formation and adhesion, followed by OEO and thymol. These findings underscore the potential of terpene compounds as promising agents to combat persistent infections and overcome antibiotic resistance associated with *A. baumannii* biofilms. Further research is warranted to fully elucidate the terpene’s mode of action and optimize the clinical application of these compounds in combating biofilm-related infections.

## Figures and Tables

**Figure 1 antibiotics-12-01539-f001:**
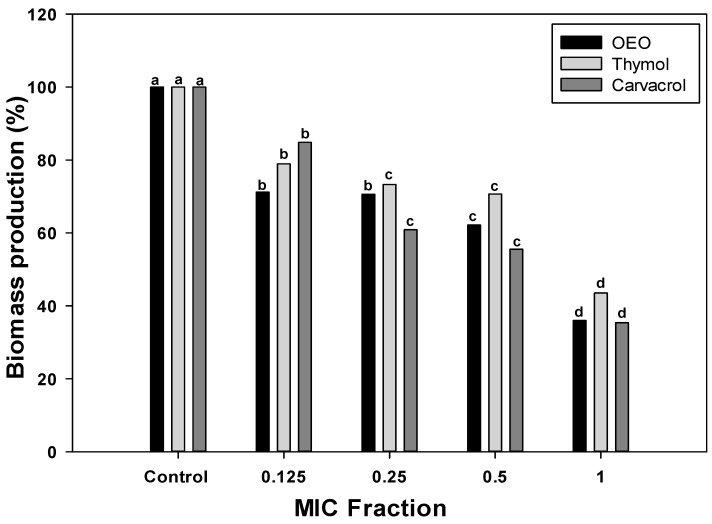
*A. baumannii* biomass reduction by OEO, thymol, and carvacrol treatments at different concentrations. Different letters among bars indicated significant differences between doses (*p* ≤ 0.05).

**Figure 2 antibiotics-12-01539-f002:**
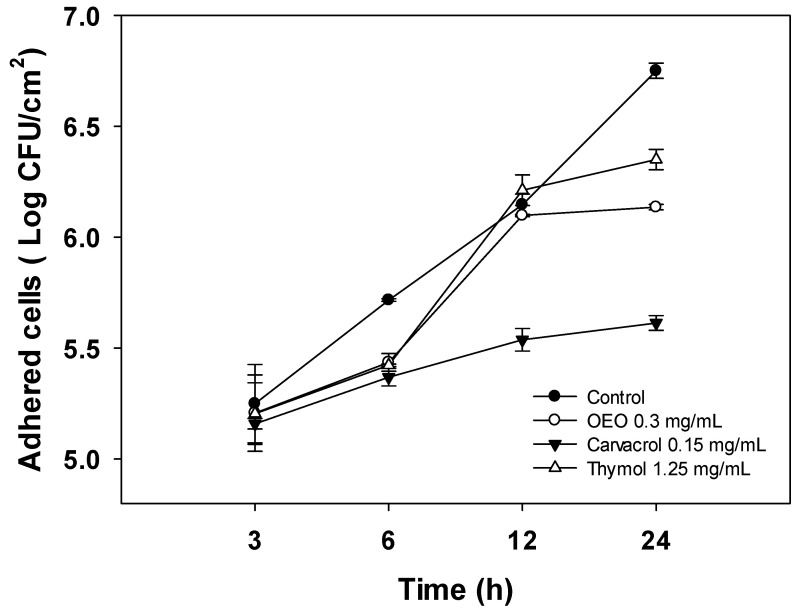
Effect of 0.5 × MIC concentrations of OEO, carvacrol, and thymol on *A. baumannii* cell density during biofilm formation on stainless steel surfaces incubated at 37 °C for 24 h. Values are expressed as the mean ± standard deviation.

**Figure 3 antibiotics-12-01539-f003:**
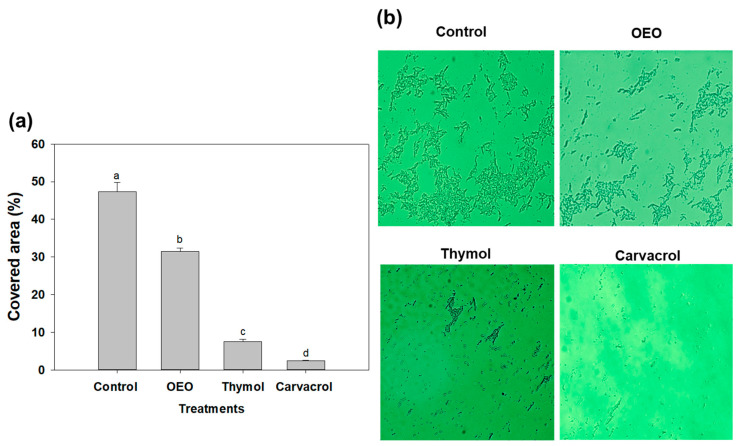
(**a**) Covered area of *A. baumannii* biofilm development in the absence of treatments (control) and exposed to 0.5 × MIC doses (OEO: 0.3 mg/mL, thymol: 1.25 mg/mL and carvacrol 0.15 mg/mL). Different letters among bars indicated significant differences (*p* ≤ 0.05). (**b**) Fluorescence microscopy of *A. baumannii* biofilm development exposed to OEO, thymol, and carvacrol on glass coverslips during incubation at 37 °C.

**Figure 4 antibiotics-12-01539-f004:**
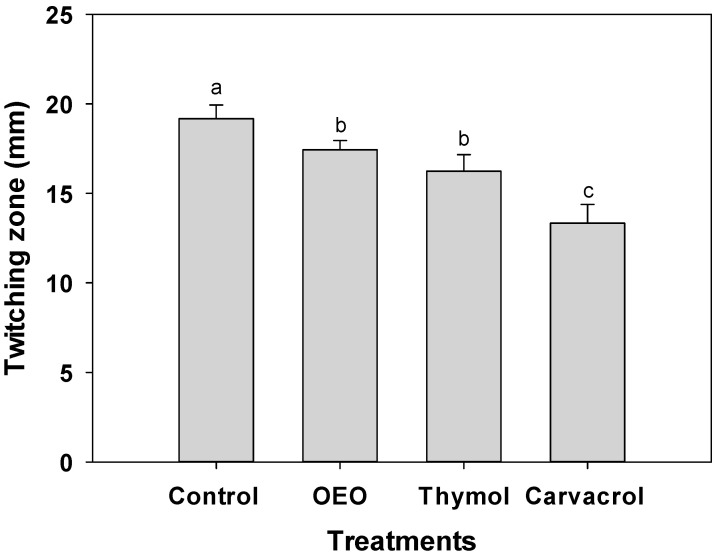
Twitching motility of *A. baumannii* exposed to 0.5 × MIC values of OEO (0.3 mg/mL), carvacrol (0.15 mg/mL), and thymol (1.25 mg/mL) incubated at 37 °C for 24 h. Values are expressed as the mean ± standard deviation (SD) of three samples. Different literals indicate significant differences (*p* ≤ 0.05).

**Table 1 antibiotics-12-01539-t001:** Minimum inhibitory (MIC) and bactericidal concentration (MBC) of OEO, carvacrol, and thymol (mg/mL) against *A. baumannii*.

*A. baumannii*	MIC (mg/mL)	MBC (mg/mL)
OEO	0.6	1.2
Carvacrol	0.3	0.6
Thymol	2.5	5

## Data Availability

The data presented in this study are available on request.

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
