# Peer review of "Inhibition of Acinetobacter baumannii Biofilm Formation by Terpenes from Oregano (Lippia graveolens) Essential Oil"

_antibiotics, 2023, doi:10.3390/antibiotics12101539_

Round 1

Reviewer 1 Report

In the Abstract:

…These compounds showed a minimum inhibitory concentration of 0.6, 0.3, and 2.5 mg/mL and a minimum bactericidal concentration of 1.2, 0.6, and 5 mg/mL, respectively, Which compounds? OEO is not a compound.

In MMs, the sentence: The tested doses exhibited remarkable efficacy in inhibiting bacterial growth, 94 with MIC values of 0.6, 0.3, and 2.5 mg/mL for OEO, carvacrol, and thymol…. I think that lacks the term respectivly at the end of the sentence.

The sentence: …the terpenoids eugenol and geraniol. Eugenol is not a terpene. In other paragraphs there ae the same mistake. Please, it is

terpenes geranyl… What geranyl? Acetate?

In the sentence:

A recent study investigated the effect of carvacrol nanoparticles… I think that the correct form is chitosan loaded essential oils.

After consulting the reference 27, an article of the authors of these manuscript, it  was possible to know that the two main compounds of the OEO were carvacrol and p-cymene, and not thymol. It would be preferable to check carvacrol and p-cymene instead thymol despite being isomers.

So, in the introduction, the sentnce: Carvacrol and thymol, the major constituents of OEO, is not correct because the OEO used by the authors had as major compounds carvacrol and p-cymene and not thymol. The reference is a review article. So, the authors should consider the main compounds of their OEO. So, I consider that the manuscript must be rejected because is based on a wrong premisse.

Author Response

Dear reviewer, thanks for your comments. This manuscript was improved according to your suggestions, and some actual bibliographical references have been added.

Reviewer 1

In the Abstract:

…These compounds showed a minimum inhibitory concentration of 0.6, 0.3, and 2.5 mg/mL and a minimum bactericidal concentration of 1.2, 0.6, and 5 mg/mL, respectively, Which compounds? OEO is not a compound.

Thanks for your comments; please find the improved text in lines 20-22.

In MMs, the sentence: The tested doses exhibited remarkable efficacy in inhibiting bacterial growth, 94 with MIC values of 0.6, 0.3, and 2.5 mg/mL for OEO, carvacrol, and thymol…. I think that lacks the term respectively at the end of the sentence.

Thanks for your comments; please find the improved text in lines 97-98.

The sentence: …the terpenoids eugenol and geraniol. Eugenol is not a terpene in other paragraphs, there are the same mistakes. Please, it is terpenes geranyl… What geranyl? Acetate?

We appreciate your observation. The authors agree, and the term “terpenoids” in the refereed sentences was replaced by “volatile compounds.” This change was also made in other paragraphs. In addition, the complete name of geranyl acetate was included. 

In the sentence:

A recent study investigated the effect of carvacrol nanoparticles… I think that the correct form is chitosan loaded essential oils.

Thanks for your comments; the text was improved in lines 254-256.

After consulting the reference 27, an article of the authors of this manuscript, it was possible to know that the two main compounds of the OEO were carvacrol and p-cymene, and not thymol. It would be preferable to check carvacrol and p-cymene instead thymol despite being isomers.

So, in the introduction, the sentence: Carvacrol and thymol, the major constituents of OEO, is not correct because the OEO used by the authors had as major compounds carvacrol and p-cymene and not thymol. The reference is a review article. So, the authors should consider the main compounds of their OEO. So, I consider that the manuscript must be rejected because it is based on a wrong premise.

We appreciate your observation. The authors agree, and the term “major constituents” in the refereed sentences was replaced by “terpenes compounds.” This change was also made in other paragraphs.

Reviewer 2 Report

Carvacrol is an active component of antibacteria and antibiofilm. In addition to oregano (Lippiagravolens), which contains large amounts of carvacrol, many medical plants also contain carvacrol. Can you add this information your introduction?

The MIC was determined as the lowest concentration that completely inhibits the visible growth of planktonic cells. To determine the MBC, the cultures from the wells that showed MIC, and the next three higher concentrations of OEO, thymol, and carvacrol were plated on LB agar and incubated at 37 °C for 24 hours.

The above definitions explain MIC and MBC, and please explain the differences in their representative significance in bacterial pharmacophysiology.

In this study, carvacrol demonstrated the lowest MIC and an MBC value twofold higher than the MIC required for inhibiting bacterial growth. (rewrite the sentence)

Your research aims to evaluate the effects of natural compounds on biofilm development. You have evidences of repression. Can you draw a schematic diagram depicting the point of inhibition of biofilm formation to make it easier to understand for readers?

No comment

Author Response

Dear reviewer, thanks for your comments. This manuscript was improved according to your suggestions, and some actual bibliographical references have been added.

Reviewer 2

Carvacrol is an active component of antibacteria and antibiofilm. In addition to oregano (Lippia gravolens), which contains large amounts of carvacrol, many medical plants also contain carvacrol. Can you add this information your introduction?

This information was added. Please find this sentence in lines 72-76.

The MIC was determined as the lowest concentration that completely inhibits the visible growth of planktonic cells. To determine the MBC, the cultures from the wells that showed MIC, and the next three higher concentrations of OEO, thymol, and carvacrol were plated on LB agar and incubated at 37 °C for 24 hours.

The above definitions explain MIC and MBC, and please explain the differences in their representative significance in bacterial pharmacophysiology.

This information was added; please find these sentences in lines 102-114.

In this study, carvacrol demonstrated the lowest MIC and an MBC value twofold higher than the MIC required for inhibiting bacterial growth. (Rewrite the sentence)

The sentence was rewritten; please find it in lines 121-122.

Your research aims to evaluate the effects of natural compounds on biofilm development. You have evidence of repression. Can you draw a schematic diagram depicting the point of inhibition of biofilm formation to make it easier to understand for readers?

We appreciate your observation. The authors agree and include a graphical abstract with A. baumannii biofilm inhibition by terpene compounds.

Round 2

Reviewer 1 Report

The authors answered the questions, although the choice of volatile compounds for the test is not clear since the main components of the essential oil were carvacrol and p-cymene and not thymol. Why did the authors choose thymol and not p-cymne? Or Why did not the authors add p-cymene in the assay?

Author Response

Response to reviewers:

The authors answered the questions, although the choice of volatile compounds for the test is not clear since the main components of the essential oil were carvacrol and p-cymene and not thymol. Why did the authors choose thymol and not p-cymene? Or why did not the authors add p-cymene in the assay?

We greatly appreciate your feedback. We decided to investigate thymol, even though p-cymene is our oregano essential oil's second most abundant constituent, because thymol is a prominent OEO compound widely recognized for its documented antimicrobial properties. Furthermore, it has been observed that monoterpenes with oxygenated groups exhibit greater inhibitory activity against bacteria compared to hydrocarbon monoterpenes like p-cymene. However, in future studies, it would be interesting to explore the effect of p-cymene on the growth of bacteria in both planktonic and biofilm states of A. baumannii. Please find this statement in lines 72-76 and 97-104.